# UniQueue

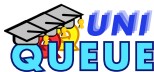

## Innowacyjny system planowania wizyt i zarządzania kolejkami dla dziekanatu Wydziału Informatyki i Telekomunikacji Politechniki Wrocławskiej

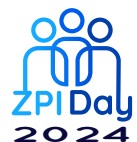

**Autorzy**: Dawid Glinkowski⁰ · Szymon Kielich⁰ · Rafał Bożek⁰ · Jakub Seredyński⁰

**Opiekun:** mgr inż. Krzysztof Stępniak

**Streszczenie**

Celem projektu UniQueue było opracowanie alternatywnego, w pełni zdalnego systemu zarządzania kolejkami dla dziekanatu Wydziału Informatyki i Telekomunikacji Politechniki Wrocławskiej. System umożliwia studentom rezerwację wizyt na konkretną godzinę, zapisywanie się do dynamicznej kolejki, otrzymywanie powiadomień o jej aktualnym stanie oraz przewidywanym czasie wizyty. W ramach projektu zaimplementowano również intuicyjny interfejs dla pracowników dziekanatu oraz panel administracyjny, który pozwala na tworzenie raportów ze statystykami i zarządzanie systemem. Opracowany został także przystępny widok aktualnego stanu stanowisk i nadchodzących wizyt na wyświetlaczu przed dziekanatem. System minimalizuje czas oczekiwania, efektywnie zarządza ruchem oraz dostarcza aktualne informacje, co zwiększa efektywność komunikacji ze studentami. Projekt został dostosowany do wymagań uczelni, uwzględniając kwestie bezpieczeństwa i ochrony danych, a także integrację z istniejącą infrastrukturą uczelnianą, zapewniając możliwość dalszego rozwoju w kolejnych iteracjach. Wykorzystano nowoczesne technologie takie jak Kotlin z frameworkiem Spring oraz React.js z TypeScript. Projekt spotkał się z pozytywnym odbiorem ze strony interesariuszy, w tym Pani Kierownik Dziekanatu oraz osób odpowiedzialnych za wsparcie informatyczne na wydziale W4.

## 1 WSTĘP

Dziekanat na uczelni wyższej pełni kluczową rolę w procesach administracyjnych związanych z obsługą studentów, takich jak odbiór dokumentów, przedłużenie ważności legitymacji studenckich czy składanie podań [12]. Aktualny system obsługi, oparty na fizycznym oczekiwaniu w kolejce przed wejściem, jest mało efektywny, co prowadzi do opóźnień i frustracji zarówno wśród studentów, jak i pracowników, a także wpływa na ruch pieszych w holu budynku C1. Dodatkowo, brak jasnej informacji o stanie kolejki oraz przewidywanym czasie oczekiwania utrudnia planowanie wizyt i efektywne zarządzanie czasem. Postępująca digitalizacja procesów administracyjnych w instytucjach edukacyjnych stawia przed uczelniami wyższymi wyzwania związane z usprawnieniem obsługi studentów.

Efektywne zarządzanie ruchem petentów w dziekanatach ma kluczowe znaczenie dla poprawy jakości usług oraz satysfakcji studentów, [16]. W dobie rosnących oczekiwań wobec dostępności i szybkości realizacji usług administracyjnych, tradycyjne systemy obsługi stają się niewystarczające.

Projekt UniQueue ma na celu zautomatyzowanie procesów związanych z obsługą studentów, skrócenie czasu oczekiwania oraz zwiększenie efektywności pracy dziekanatu poprzez wdrożenie nowoczesnego systemu planowania wizyt i zarządzania kolejkami. Eliminuje on również konieczność korzystania z druku termotransferowego, co przyczynia się do większej dbałości o ekologię.

### 1.1 Cele projektu

Jednym z celów projektu UniQueue było usprawnienie procesu obsługi studenckiej poprzez wprowadzenie:

- **Rezerwacji wizyt online**: Wprowadzenie możliwości rezerwacji wizyt na konkretną godzinę, uwzględniając:
    - Godziny otwarcia dziekanatu (regularne i nieregularne),
    - Przerwy tymczasowe i cykliczne,
    - Zarezerwowane już terminy,
    - Estymowany czas trwania wizyty.

- **Dynamicznej kolejki biletowej**: Umożliwienie studentom bez rezerwacji zapisanie się do dodatkowej kolejki biletowej, z priorytetyzacją wizyt umówionych.

- **Cyfrowych tokenów**: Zastąpienie papierowych biletów cyfrowymi tokenami, eliminując konieczność fizycznego oczekiwania przed dziekanatem oraz minimalizując koszty druku.

- **Zdalnych powiadomień**: Opracowanie modułu powiadomień o zbliżającej się wizycie poprzez e-mail oraz możliwość eksportu terminu do kalendarza.

- **Integracji z systemami uczelnianymi**: Autoryzacja keycloak kontami AD wykorzystywanymi również w USOS/Active Directory, zapewniając jednolite logowanie i zgodność z polityką bezpieczeństwa uczelni [5].

- **Bezpieczeństwa danych**: Minimalizacja przetwarzanych danych osobowych zgodnie z RODO, przetwarzając jedynie niezbędne informacje, czyli adres e-mail studenta [11].

- **Dostępności**: Zwiększenie dostępności systemu, w szczególności dla osób słabowidzących.

Projekt miał na celu nie tylko techniczne usprawnienie procesu, ale również zwiększenie zadowolenia użytkowników końcowych poprzez intuicyjny interfejs i łatwość obsługi, aby nie powodować dodatkowego obciążenia dla osób przyzwyczajonych do aktualnych rozwiązań [4].

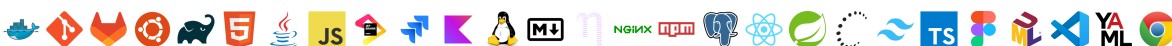

Rysunek 1: Technologie użyte w projekcie

## 1.2 Wybrane technologie

Do realizacji projektu wybrano następujące technologie:

- **Backend**: Kotlin z wykorzystaniem Spring Framework, co zapewnia wydajność, skalowalność aplikacji oraz zgodność ze standardami branżowymi [7], [6].

- **Frontend**: React.js z TypeScript oraz Tailwind CSS, co umożliwia tworzenie nowoczesnych, responsywnych i dostępnych interfejsów użytkownika [8], [9].

- **Baza danych**: PostgreSQL, oferująca zaawansowane możliwości przechowywania i zarządzania danymi oraz niezawodność [10].

- **Autoryzacja**: Keycloak z protokołem OpenID Connect, zapewniający bezpieczne logowanie użytkowników i integrację z systemem SSO uczelni [5].

- **Wyświetlanie informacji**: System Samsung MagicInfo na monitorze, umożliwiający centralne zarządzanie treściami wyświetlanymi na ekranach.

## 2 PRACE ZWIĄZANE Z TEMATEM

Przed przystąpieniem do realizacji projektu przeprowadzono analizę istniejących rozwiązań zarządzania kolejkami oraz literatury z zakresu teorii kolejek i zarządzania ruchem petentów, [2]. Stwierdzono, że większość dostępnych systemów opiera się na statycznych rezerwacjach, nie uwzględniając dynamicznych zmian, takich jak opóźnienia, anulowanie wizyt czy zmienne obciążenie stanowisk. W odpowiedzi na te ograniczenia, UniQueue został zaprojektowany jako system dynamiczny, integrujący się z istniejącymi systemami uczelnianymi oraz dostosowujący się do bieżących warunków.

W ramach prac projektowych odbyły się liczne konsultacje z pracownikami dziekanatu w celu zidentyfikowania kluczowych potrzeb i wymagań. Na podstawie zebranych informacji ustalono priorytety funkcjonalności oraz dokonano wyboru technologii, które pozwolą na osiągnięcie zamierzonych celów [14].

Integracja z systemem Keycloak, wykorzystywanym przez Politechnikę Wrocławską jako metoda jednokrotnego logowania (SSO), umożliwia bezproblemowe włączenie UniQueue w istniejącą infrastrukturę uczelnianą [5]. Dzięki temu studenci mogą korzystać z systemu bez konieczności tworzenia dodatkowych kont, co zwiększa wygodę i bezpieczeństwo.

Na podstawie przeprowadzonego wywiadu wśród studentów stwierdzono, że zainteresowanie dedykowaną aplikacją mobilną jest niewielkie. W związku z tym zdecydowano się na zbudowanie responsywnej strony internetowej dostępnej zarówno na urządzeniach mobilnych, jak i stacjonarnych, co pozwala

na zapewnienie dostępności dla wszystkich użytkowników z wielu platform. W celu wczesnej weryfikacji oczekiwań i wymagań, do stworzenia makiet interfejsów wykorzystano narzędzie Figma, co umożliwiło efektywne zbieranie opinii i dostosowywanie projektu do potrzeb użytkowników.

# 3 WYNIKI

## 3.1 Zaimplementowane funkcjonalności

System UniQueue składa się z czterech głównych interfejsów:

### 3.1.1 Interfejs dla studentów

- **Logowanie**: Możliwość logowania przez Active Directory uczelni lub wysyłanie próśb o wizytę dostarczaną bezpośrednio do pracownika dziekanatu poprzez formularz dla osób spoza uczelni.

- **Wybór typu wizyty**: Opcja zapisania się na wizytę na konkretną godzinę lub do dynamicznej kolejki biletowej.

- **Wybór tematu wizyty**: Na podstawie wybranego tematu system filtruje dostępne stanowiska obsługujące dany temat.

- **Zmiana języka**: Funkcjonalność zmiany języka interfejsu, z preferencjami zapisywanymi w bazie danych. Ustawienia językowe są propagowane do innych komponentów, takich jak eksportowane pliki ICS do kalendarza czy wybór odpowiedniego szablonu e-mail.

- **Responsywność i dostępność**: Strona jest skalowalna i responsywna, dostosowuje się do różnych urządzeń dzięki zaimplementowanemu mechanizmowi detekcji typu urządzenia.

- **Zarządzanie wizytami**: Na ekranie głównym wyświetlane są karty reprezentujące wizyty, którymi można zarządzać. W przypadku, gdy student jest już zapisany do kolejki, system uniemożliwia ponowne zapisanie się, prezentując informacje o aktualnej wizycie.

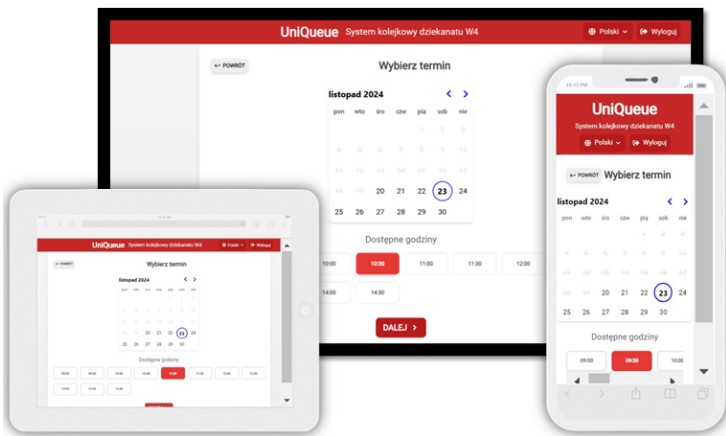

Rysunek 2: Interfejs dla studentów

### 3.1.2 Panel administracyjny

- **Zarządzanie stanowiskami**: Operacje CRUD (Create, Read, Update, Delete) dla stanowisk, w tym aktywacja, dezaktywacja oraz ukrywanie ich na ekranach dostępnych dla studentów.

- **Statystyki**: Moduł umożliwiający analizę danych dotyczących wizyt, obciążenia stanowisk oraz efektywności działania systemu, np. średnia dzienna liczba wizyt na stanowisko i średni czas trwania wizyty.

- **Zarządzanie tematami wizyt**: Definiowanie szczegółów tematów oraz przypisywanie ich do stanowisk.

- **Zarządzanie pracownikami**: Przypisywanie ról i uprawnień pracownikom korzystającym z systemu.

- **Konfiguracja systemu**: Ustalanie godzin otwarcia, przerw tymczasowych i cyklicznych.

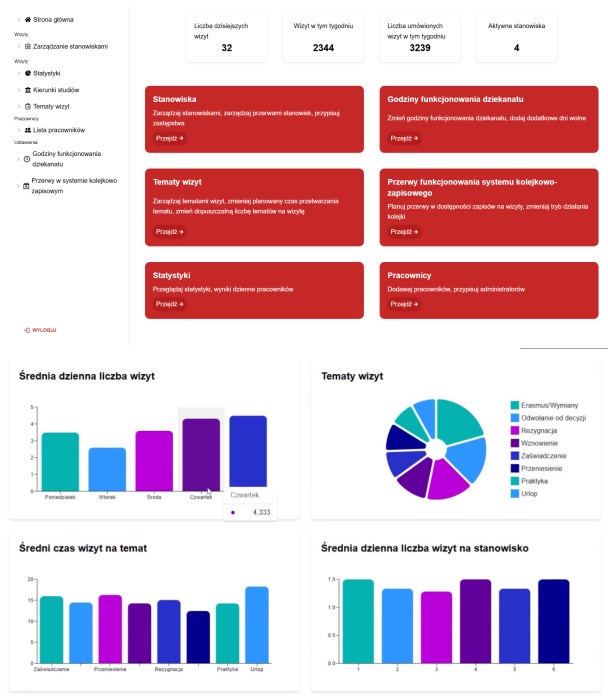

Rysunek 3: Panel administracyjny

### 3.1.3 Interfejs dla pracowników

- **Obsługa wizyt**: Rozpoczynanie, kończenie i pomijanie wizyt, ręczne dodawanie wizyt dla osób spoza uczelni.

- **Zarządzanie stanowiskiem**: Otwieranie i zamykanie stanowiska, planowanie przerw oraz przekierowywanie spraw do innych okienek.

- **Podgląd kolejki**: Monitorowanie stanu kolejki i zarządzanie jej przebiegiem w czasie rzeczywistym poprzez zapraszanie konkretnych petentów.

- **PWA**: umożliwia pobranie przypięcie do paska startowego i zapobiega przypadkowym zamknięciom.

- **Intuicyjny interfejs**: Interfejs został uproszczony, aby nie powodować dodatkowego obciążenia dla pracowników przyzwyczajonych do poprzedniego systemu. [4]

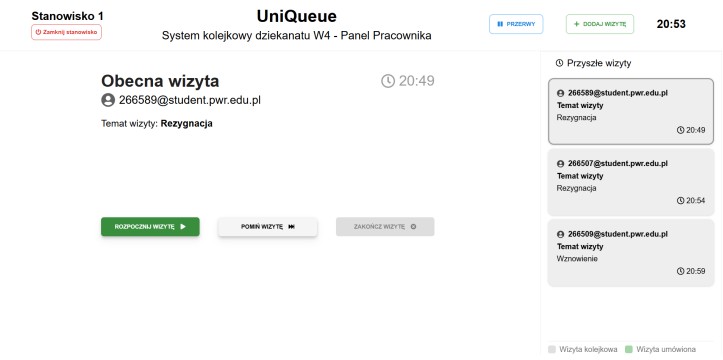

Rysunek 4: Interfejs dla pracowników

### 3.1.4 Interfejs na monitor

- **Wyświetlanie stanu kolejki**: Informacje o aktualnym stanie kolejek, nadchodzących wizytach oraz komunikaty dla studentów. Dane są aktualizowane w czasie rzeczywistym dzięki komunikacji przez WebSockety.

- **Moduł informacyjny**: Wyświetlanie treści informacyjnych oraz dostarczenie kodu QR do strony systemu w widocznym miejscu, co ułatwia i przyspiesza dotarcie do strony.

- **Dostępność**: Dostosowanie przystępności dla osób słabowidzących poprzez zwiększony kontrast i czytelność, co jest wymagane w przestrzeni publicznej.

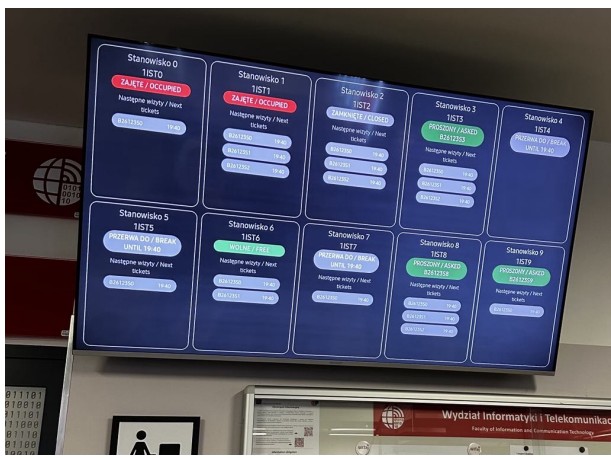

Rysunek 5: Monitor przed dziekanatem

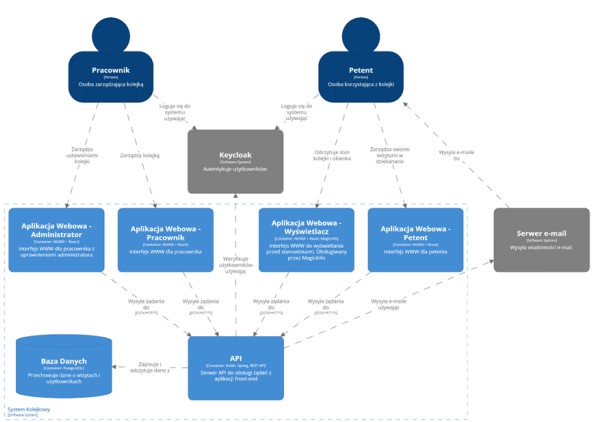

Rysunek 6: Diagram architektury C4

## 3.2 Optymalizacje i wyzwania

Podczas realizacji projektu zespół musiał sprostać różnym wyzwaniom:

- **Algorytmika**: Implementacja efektywnych algorytmów kolejkowania wizyt [1] do łączenia i znajdowania wolnych okien czasowych. Pierwszym krokiem w algorytmie jest określenie przedziałów czasu, w których stanowisko jest dostępne. Aby wyliczyć te przedziały trzeba najpierw podać okresy, które określają kiedy okienko jest czynne i odjąć od nich wszystkie przerwy, oraz inne wizyty. Można to wykonać w sposób naiwny, poprzez porównanie każdego dostępnego okresu z każdą przerwa, jednak złożoność takiego rozwiązania jest kwadratowa, czyli nieefektywna. Zastosowanym rozwiązaniem przez nas jest wpierw posortowanie osobno dostępnych godzin, oraz przerw według ich początków, oraz złączenie tych przedziałów które na siebie nachodzą i oznaczają to samo - przerwy z przerwami, godziny otwarcia z godzinami otwarcia. Poprzedni krok pozwala na porównywanie tylko tych dostępnych godzin i przerw, które na siebie nachodzą. Złożoność wyliczenia efektywnych godzin otwarcia wtedy jest liniowa, jednak wcześniej odbyło się sortowanie, więc końcowo złożoność wynosi $O(n \log n + m \log m)$, gdzie $n$ to liczba przedziałów otwarcia, a $m$ to liczba przerw.

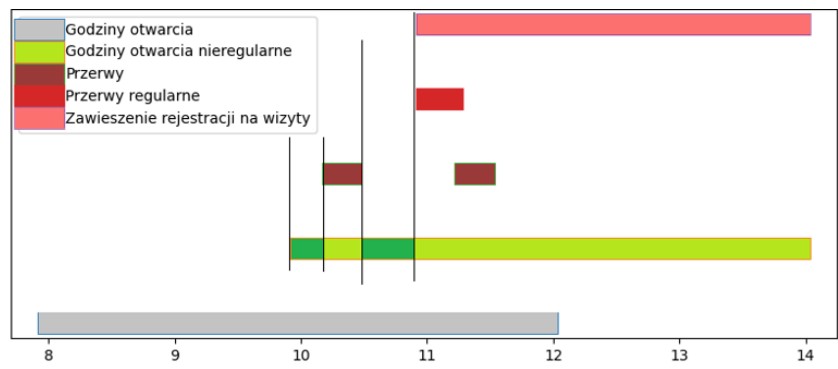

Rysunek 7: Wizualizacja problemu łączenia przedziałów

- **Zmieniające się wymagania**: Przykładowo, początkowo zastosowano mechanizm autoryzacji mailowej dla osób spoza uczelni. Jednakże, na skutek uwag zgłoszonych przez interesariuszy, podjęto decyzję o zamknięciu tego rozwiązania i wprowadzeniu alternatywy w postaci formularza kontaktowego.

- **Ograniczenia infrastrukturalne**: Konieczność działania na starszych bibliotekach i użycia polyfilli ze względu na ograniczenia sprzętowe uczelni. Przykładowo, przeglądarka w playerze monitora obsługiwała jedynie standardy webowe do roku 2012, co wymagało zastosowania odpowiednich technologii [15].

- **Bezpieczeństwo**: Minimalizacja przetwarzanych danych osobowych do adresu e-mail oraz zgodność z RODO. Przeprowadzone zostaną audyty bezpieczeństwa i testy penetracyjne, potwierdzające skuteczność mechanizmów ochrony danych [13].

- **Cache'owanie**: Przerwy z bieżącej daty są zapisywane w pamięci podręcznej (cache), który aktualizuje się w przypadku zmian w tabeli przerw, co przyspiesza dostęp do danych i odciąża bazę danych.

- **Kompatybilność przeglądarek**: Wykorzystanie PostCSS i zastąpienie nowoczesnych rozwiązań CSS starszymi odpowiednikami, co zapewniło kompatybilność z przeglądarkami o ograniczonej funkcjonalności [15].

- **Testowanie**: W każdym z czterech frontendów minimalne pokrycie testami to 80% linii kodu, co zapewnia niezawodność systemu i ułatwia wykrywanie błędów.

## 3.3 Integracje i wdrożenie

System został zintegrowany z istniejącą infrastrukturą uczelni:

- **Autoryzacja**: Aktualnie korzysta ze skonfigurowanego przez nas serwera autoryzacyjnego Keycloak, przygotowanego pod integrację z faktycznym systemem uczelnianym. Pozwala to na łatwe przejście do produkcyjnego środowiska uczelni.

- **Wyświetlanie informacji**: Integracja z systemem Samsung MagicInfo na monitorze, umożliwiającym centralne zarządzanie treściami i dostosowanie wyświetlanych informacji do potrzeb dziekanatu.

- **Wdrożenie**: System wdrożono na serwerze testowym, zabezpieczonym certyfikatem HTTPS, co zapewnia bezpieczną komunikację i zgodność z wymaganiami bezpieczeństwa uczelni.

- **Repozytorium**: Kod źródłowy zostanie umieszczony w prywatnym repozytorium uczelni, zgodnie z polityką bezpieczeństwa i procedurami uczelnianymi.

# 4 WNIOSKI

## 4.1 Podsumowanie

Projekt UniQueue zrealizował założone cele, tworząc zintegrowany i nowoczesny system zarządzania kolejkami dostosowany do specyficznych potrzeb dziekanatu Wydziału Informatyki i Telekomunikacji. System ten nie tylko zwiększa efektywność obsługi studentów i skraca czas oczekiwania, ale także poprawia komunikację pomiędzy dziekanatem a studentami oraz usprawnia pracę personelu administracyjnego.

Osiągnięte rezultaty potwierdzają, że wykorzystanie nowoczesnych technologii oraz bliska współpraca z użytkownikami końcowymi prowadzi do powstania rozwiązań spełniających realne potrzeby i oczekiwania. Projekt spotkał się z pozytywnym odbiorem ze strony interesariuszy, co świadczy o jego potencjale wdrożeniowym.

## 4.2 Znaczenie i przyszłe implikacje

UniQueue stanowi przykład udanego zastosowania nowoczesnych technologii informatycznych w sektorze edukacji wyższej. Wdrożenie systemu może przyczynić się do podniesienia jakości obsługi studentów, zwiększenia satysfakcji z usług administracyjnych oraz optymalizacji procesów wewnętrznych uczelni.

Ponadto, doświadczenia zdobyte podczas realizacji projektu mogą być wykorzystane przy opracowywaniu podobnych systemów w innych jednostkach uczelni lub w innych instytucjach publicznych. UniQueue może stać się inspiracją dla dalszych działań w kierunku cyfryzacji i automatyzacji procesów administracyjnych.

## 4.3 Kierunki rozwoju

Możliwe kierunki dalszego rozwoju projektu obejmują:

- **Uczenie maszynowe**: Wykorzystanie algorytmów uczenia maszynowego do przewidywania czasu obsługi i automatycznej optymalizacji kolejek w zależności od natężenia ruchu [3].

- **Rozszerzenie modułu powiadomień**: Wprowadzenie interaktywnych powiadomień SMS lub poprzez aplikację mobilną, co zwiększy dostępność informacji dla użytkowników.

- **Integracja z systemem USOS**: Automatyczne pobieranie danych studentów i harmonogramów zajęć, co usprawni proces rezerwacji wizyt i pozwoli na lepsze dostosowanie terminów.

- **Analiza danych**: Zaawansowane narzędzia analityczne do monitorowania wydajności systemu, satysfakcji użytkowników oraz identyfikacji obszarów do poprawy, umożliwiające ciągłe doskonalenie usług.

- **Integracja z kalendarzem**: Automatyczne dodawanie wizyt do kalendarza uczelnianego.

## 4.4 Podziękowania

Dziękujemy naszemu promotorowi, mgr inż. Krzysztofowi Stępniakowi, za cenne wskazówki przy konfiguracji testowego serwera autoryzacyjnego Keycloak, za umożliwienie testowania na monitorze z systemem MagicInfo oraz wsparcie w uzyskaniu certyfikatów HTTPS i domeny. Dziękujemy również zespołowi bezpieczeństwa informacji za możliwość przeprowadzenia testów naszej aplikacji oraz pracownikom dziekanatu za konstruktywne uwagi w trakcie realizacji projektu.

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
