# OpenReview forum: "UniQueue"
_pwr.edu.pl/Wrocław_University_of_Science_and_Technology/2024/ZPI_Day — Wrocław University of Science and Technology 2024 ZPI Day Submission_

### Official Review · Reviewer_2ky3 · 2024-12-03
**Oceniany projekt UniQueue charakteryzuje się dużym potencjałem implementacyjnym dla wielu dziekanatów uczelni wyższych. Przedstawiona funkcjonalność systemu  ma również potencjał rozwojowy.**

**Confidence:** 4
**Significance Of Results:** 4
**Overall Quality:** 4

**Compliance With Template:**

4: High Quality – The article contains all the required sections, which are well-written and substantively correct, although minor errors or shortcomings may be present. The overall structure is clear and coherent.

**Description Of Results:**

4: High Quality – The results are described in detail and supported by usage examples or evaluations. The description is reliable but may lack full depth of analysis.

**Feedback On Consistency:**

Sam sposób prezentacji wydaje się spójny i logiczny. Pozytywnie wpływa na zrozumienie jego działania i zastosowania duża liczba rysunków/fotografii.
Drobną niedoskonałością może być zbyt krótki i lakoniczny opis optymalizacji i algorytmów wykorzystanych w systemie.

**Potential For Development:**

Jest bardzo duży potencjał zarówno w optymalizacji, zwiększeniu funkcjonalności jak i integracji.
Szczególnie ciekawa wydaje się integracja z systemem USOS.

**Project Nature Evaluation:**

Pod względem inżynierskim projekt oceniam wysoko. Jest tu zarówno opracowana aplikacja klienta jak i administratora, obie są połączone z bazą danych. Jest możliwość wyświetlania na monitorze przed dziekanatem.  Jest również zaimplementowana responsywność systemu. Są raporty.

**Technical Language Precision:**

4: High Quality – The language is appropriate for a technical report. Terminology is used correctly, and statements are precise, with only minor shortcomings that do not affect the overall clarity.

---

### Official Review · Reviewer_yRGu · 2024-12-04
**Interesujący projekt dla uczelni**

**Confidence:** 4
**Significance Of Results:** 5
**Overall Quality:** 5

**Compliance With Template:**

5: Very High Quality – The article contains all the required sections, which are written in a very detailed, clear, and error-free manner. The structure is professional and meets expectations, and the content adheres to the highest substantive and formal standards.

**Description Of Results:**

5: Very High Quality – The results are described in detail, clearly and comprehensively, supported by thorough evaluation, analysis, and convincing usage examples. The description meets the highest substantive standards.

**Feedback On Consistency:**

Analiza projektu pod względem istniejących rozwiązań oraz wymagań użytkowników została opisana poprawnie. Wyniki prac oraz podsumowanie są logiczne i spójne z założeniami. Bardzo pozytywną stroną projektu jest dokonanie rzetelnej oceny obecnej sytuacji, włączając w to konsultacje dotyczące funkcjonalności projektu, skonfrontowane z oczekiwaniami władz dziekanatu i pracowników tej jednostki.

**Potential For Development:**

Zaproponowane rozwiazanie pozwala na wszechstronny rozwój aplikacji, ze wskazaniem na wprowadzenie zaawansowanych opcji samego interfejsu jak i szerszych statystyk.

**Project Nature Evaluation:**

W projekcie wykorzystano obecnie używane technologie, które musiały być zintegrowane ze sobą w celu poprawnej współpracy. Zaimplementowano rozwiązania pozwalające na optymalizację wyników.

**Technical Language Precision:**

5: Very High Quality – The language is entirely appropriate for a technical report. All terms are used correctly and precisely, and the style is professional, clear, and coherent, without any errors or ambiguities.

---

### Official Review · Reviewer_8xkM · 2024-12-09
**UniQueue**

**Confidence:** 5
**Significance Of Results:** 4
**Overall Quality:** 5

**Compliance With Template:**

5: Very High Quality – The article contains all the required sections, which are written in a very detailed, clear, and error-free manner. The structure is professional and meets expectations, and the content adheres to the highest substantive and formal standards.

**Description Of Results:**

5: Very High Quality – The results are described in detail, clearly and comprehensively, supported by thorough evaluation, analysis, and convincing usage examples. The description meets the highest substantive standards.

**Feedback On Consistency:**

EN:
The project description is consistent and well-structured, effectively transitioning from problem analysis to solution implementation and results. The challenges of the existing queuing system are clearly articulated, and the proposed solutions align logically with these challenges. The presentation of results is detailed, emphasizing the platform's technical functionalities and its positive reception by stakeholders. However, the conclusions could benefit from a more explicit connection between the results and the stated objectives, particularly regarding quantitative improvements in user experience or operational efficiency.

PL:
Opis projektu jest spójny i dobrze skonstruowany, skutecznie przechodząc od analizy problemu do wdrożenia rozwiązania i wyników. Wyzwania związane z istniejącym systemem kolejkowym są jasno sformułowane, a proponowane rozwiązania logicznie do nich pasują. Prezentacja wyników jest szczegółowa, podkreślając techniczne funkcjonalności platformy i jej pozytywny odbiór przez interesariuszy. Wnioski mogłyby jednak zyskać na wyraźniejszym powiązaniu wyników z określonymi celami, szczególnie w odniesieniu do ilościowej poprawy doświadczenia użytkownika lub wydajności operacyjnej.

**Potential For Development:**

EN:
The article identifies several promising directions for future development, including:
Machine Learning - predictive models for optimizing queue lengths and reducing wait times,
Enhanced Notification Systems - integration of SMS and push notifications to improve communication.
Mobile Application - dedicated iOS and Android apps for increased accessibility and user engagement.
Advanced Analytics - tools for monitoring system efficiency, user behavior, and satisfaction.
While these proposals are promising, the article could benefit from providing a concrete roadmap or feasibility analysis for these enhancements. Detailed implementation plans or timelines would strengthen the argument for the platform's scalability and long-term applicability.

PL:
W artykule zidentyfikowano kilka obiecujących kierunków przyszłego rozwoju, w tym:
- uczenie maszynowe - modele predykcyjne do optymalizacji długości kolejek i skracania czasu oczekiwania,
- ulepszone systemy powiadomień - integracja powiadomień SMS i push w celu poprawy komunikacji.
- aplikacje mobilne - dedykowane aplikacje na systemy iOS i Android w celu zwiększenia dostępności i zaangażowania użytkowników.
- zaawansowana analityka - narzędzia do monitorowania wydajności systemu, zachowań użytkowników i ich satysfakcji.
Chociaż propozycje te są obiecujące, artykuł mógłby skorzystać na przedstawieniu konkretnej mapy drogowej lub analizy wykonalności tych ulepszeń. Szczegółowe plany wdrożenia lub harmonogramy wzmocniłyby argumenty za skalowalnością platformy i jej długoterminowym zastosowaniem.

**Project Nature Evaluation:**

EN:
The project exhibits strong engineering characteristics, showcasing the application of modern technologies such as Kotlin, React.js, and PostgreSQL. Its focus on system scalability, efficient algorithms, and integration with the university’s existing infrastructure reflects thoughtful engineering design. The use of secure authentication methods and compliance with GDPR further highlight its technical rigor. While the system meets specific institutional needs, its broader utility and adaptability for different contexts (e.g., other public service domains) remain underexplored.

PL:
Projekt wykazuje najważniejsze cechy inżynieryjne, prezentując zastosowanie nowoczesnych technologii, takich jak Kotlin, React.js i PostgreSQL. Skupienie się na skalowalności systemu, wydajnych algorytmach i integracji z istniejącą infrastrukturą uniwersytetu odzwierciedla przemyślany projekt inżynieryjny. Zastosowanie bezpiecznych metod uwierzytelniania i zgodność z RODO dodatkowo podkreślają jego rygor techniczny. Podczas gdy system spełnia określone potrzeby instytucjonalne, jego szersza użyteczność i możliwość dostosowania do innych kontekstów (np. innych domen usług publicznych) pozostaje niezbadana.

**Technical Language Precision:**

5: Very High Quality – The language is entirely appropriate for a technical report. All terms are used correctly and precisely, and the style is professional, clear, and coherent, without any errors or ambiguities.

---

### Official Review · Reviewer_YRmj · 2024-12-09
**UniQueue Innowacyjny system planowania wizyt i zarządzania kolejkami dla dziekanatu Wydziału Informatyki i Telekomunikacji Politechniki Wrocławskiej**

**Confidence:** 4
**Significance Of Results:** 4
**Overall Quality:** 4

**Compliance With Template:**

4: High Quality – The article contains all the required sections, which are well-written and substantively correct, although minor errors or shortcomings may be present. The overall structure is clear and coherent.

**Description Of Results:**

4: High Quality – The results are described in detail and supported by usage examples or evaluations. The description is reliable but may lack full depth of analysis.

**Feedback On Consistency:**

Opis projektu jest spójny, a prezentowane informacje są w większości logicznie powiązane. Przydatne byłoby syntetyczne porównanie możliwości istniejących rozwiązań wraz z porównaniem z proponowanym w postaci przejrzystej tabeli.
Zauważalna jest słaba przejrzystość rysunku nr 6. Wizualizacja problemu łączenia przedziałów, wobec zastosowania zbliżonych oznaczeń kolorowych dla trzech pozycji z legendy czytelnik może nie rozróżnić wartości, a w przypadku druku czarnobiałego rozróżnienie wartości może się stać całkowicie nieczytelne. Zamieszczenie schematu architektury systemu ułatwiłoby również zrozumienie całego opisu.

**Potential For Development:**

Artykuł wskazuje jakie aspekty projektu wymagają rozwoju w przyszłości. Zaproponowany system może stanowić narzędzie badań algorytmów harmonogramowania pracy stanowisk obsługi w tym konkretnym wycinku rzeczywistości również z metodami maszynowego uczenia jak i innych wskazanych przez autorów przestrzeniach.

**Project Nature Evaluation:**

Projekt ma wyraźny cel inżynierski, zastosowane metody oraz rozwiązania są odpowiednie do realizacji tego celu.

**Technical Language Precision:**

4: High Quality – The language is appropriate for a technical report. Terminology is used correctly, and statements are precise, with only minor shortcomings that do not affect the overall clarity.

---

### Decision · Program_Chairs · 2024-12-10

Accept (Oral)